# Impacts of Material Orthotropy on Mechanical Behaviors of Asphalt Pavements

**Miao Lin** [1,2], **Changbin Hu** [1], **Hongxin Guan** [2], **Said M. Easa** [1,3] **and Zhenliang Jiang** [1,4,*]

1   College of Civil Engineering, Fuzhou University, Fuzhou 350108, China; linmiaofj@hotmail.com (M.L.);
    huchangbin@fzu.edu.cn (C.H.); seasa@ryerson.ca (S.M.E.)
2   College of Transportation Engineering, Changsha University of Science and Technology,
    Changsha 410114, China; guanhongxincs@163.com
3   Department of Civil Engineering, Ryerson University, Toronto, ON M5B 2K3, Canada
4   Department of Civil and Environmental Engineering, The Hong Kong University of Science & Technology,
    Clear Water Bay, Kowloon, Hong Kong, China
*   Correspondence: zhenliangjiang@foxmail.com

**Abstract:** Material anisotropy significantly impacts the mechanical behaviors of asphalt pavements. However, most current asphalt pavement design methods treat the material properties only as isotropic, which could significantly skew the mechanical behaviors. There is a need to evaluate the impact of material anisotropy on pavement mechanical behaviors. In this study, we first developed a new and efficient 3-dimensional finite element (FE) model of anisotropic material. Then, the feasibility of the proposed FE model was verified using field data collected with a falling weight deflectometer. Finally, using this model, the contributions of each layer anisotropy to the mechanical properties were determined. The results showed that the mechanical behaviors were more sensitive to the orthotropy than to the transverse isotropy of the material. The all-layer orthotropy was the most unfavorable combination. In addition, the subgrade orthotropy showed the most significant effect on increasing the surface deflection and compressive strain of the subgrade top (by about 10%). Based on the study results, we recommend that the homogeneity degree of the filling subgrade should be strictly controlled to ensure adequate pavement capacity and anti-rutting performance during construction.

**Keywords:** asphalt pavements; material; orthotropy; finite element; mechanical behavior





**Highlights:**

1.   A new and efficient 3D FE model considering material orthotropy was developed.
2.   The material anisotropy 3D FE model was built using the experiment-based inputs.
3.   Software programmed by MATLAB was proposed to calculate the orthotropic parameters.
4.   The impacts of the all-layer and each layer anisotropy were thoroughly analyzed.
5.   Pavement mechanical behaviors were found to be more sensitive to the material orthotropy.
6.   The feasibility of the proposed FE model was verified using the field data.

## 1. Introduction

Asphalt pavement is a type of vertically compacted material during construction that suffers vertical traffic loads in service. Therefore, each layer of pavement material should be considered anisotropic [1–3]. An anisotropic material indicates that the material properties (the stiffness, modulus, etc.) vary in three directions, i.e., the vertical, longitudinal, and transverse [4,5]. However, most current asphalt pavement design methods only consider the material properties as isotropic, which can significantly skew the mechanical behaviors of asphalt pavements [6,7].

In general, the anisotropic properties of asphalt pavement materials include transverse isotropy and orthotropy [8–12]. The transverse isotropy of asphalt concrete (AC) was first investigated by Masad et al. [13]. After that, Wang et al. [11] verified transverse isotropy in field samples using the triaxial test. The results showed that the degree of anisotropy of the AC samples was in the range of 20%–50%. Then, the dynamic moduli of the AC samples along the vertical and longitudinal directions were measured by Motola and Ozan [14], showing that the longitudinal moduli and stiffness were 40% and 30% of that in vertical. Also, the vertical modulus and out-of-plane Poisson's ratio were higher than the longitudinal modulus and in-plane Poisson's ratio, respectively [15]. Moreover, Alanazi et al. [16] stated that the anisotropy decreased with the increasing density.

Without considering the transverse isotropy of pavement materials, an error prediction in pavement fatigue cracking and potential rutting would occur [17,18]. In addition, previous studies showed that the damage of asphalt pavement increased with the growing transverse isotropy [4]. The mechanical behaviors of asphalt pavement between transverse isotropic and isotropic materials were compared [19]. We found that, when considering the transverse isotropy, the tensile and compressive strains were higher than those when considering the material as isotropic. In addition, the vertical strain obtained by the transverse isotropy model agreed well with the measured value [20]. Islam et al. [21] demonstrated that when the degrees of anisotropy of pavement materials were 0.80 and 0.87, the FE-based vertical and horizontal stress matched well with field data.

In addition, the transverse isotropy effects of the separated layer on the pavement mechanical behaviors were discussed in several studies, and detailed information for these layers can be found in the literature, such as surface layer [22,23], semi-rigid base layer [24–26], sub-base layer [27,28], and subgrade layer [29–31]. Tarefder et al. [4] investigated the transverse isotropy effects of the combined layers, including hot-mix asphalt (HMA), base, sub-base, and subgrade, on the pavement mechanical behaviors. They argued that the transverse isotropy effects on the HMA and all-layer combination structures were significant in the pavement response.

Materials with three mutually orthogonal elastic symmetry surfaces are defined as orthotropy [32]. According to this definition, the elastic modulus, Poisson's ratio, and shear modulus of pavement material are different in the transverse and longitudinal directions. Yang et al. [33] showed that the surface deflection and tensile stress at the bottom of the semi-rigid base layer were significantly affected by the anisotropy in two directions of the horizontal plane.

There is a need to include material anisotropy in pavement design and evaluation [34]. However, most researchers only investigated the effects of transverse isotropy on pavement mechanical behaviors [12,28]. In general, the transverse isotropy is a simplified case, since the material properties are identical in the transverse and longitudinal directions [35,36]. This case does not conform to the actual anisotropic properties of asphalt pavement materials [4]. Recently, the orthotropy (i.e., different material properties in the transverse and longitudinal directions) of pavement materials was proposed and considered in the HMA layer to increase the prediction accuracy [33].

In recent decades, the finite element (FE) model has become a widely used simulation method in pavement engineering due to its universality and effectiveness [37,38]. In particular, with the help of commercial software (e.g., SAP, NASTRAN, ANSYS, and ABQUES), the pavement mechanical behaviors [39,40] (e.g., deformation, stress, and strain) at any position can be obtained easily. Therefore, the FE model has increasingly been implemented to evaluate the effect of material anisotropy on pavement mechanical behaviors [5]. It should be noted that the model inputs (material parameters), such as the elastic modulus and Poisson's ratio, can considerably impact the evaluation results. However, these inputs in most literature are based on assumptions, which may not reflect the actual pavement performance. Therefore, it is critical to determine the inputs of the FE model via the experimental data to evaluate the pavement mechanical behaviors.

To fill these gaps, the effect of material orthotropy of each layer on the mechanical behaviors was investigated, and a new and efficient method for determining material parameters was developed. Specifically, this paper aims to address two questions: What are the parameter relationships of the orthotropic body considering the consistency of the parameters and What are the impacts of material orthotropy on the mechanical behaviors of asphalt pavement? The main contributions of the present work are as follows:

(1)  We developed a new and efficient 3D FE model of the anisotropic pavement material considering the consistency of the parameters (e.g., the elastic modulus, shear modulus, and Poisson's ratio), which derives the relationship among anisotropic parameters.

(2)  Based on the degree of anisotropy and considering the parameter consistency, the anisotropic model's independent parameters were reduced from nine to five, which is convenient for determining the input parameters.

(3)  We designed nine different combinations of material characteristics considering the contributions of each layer of material characteristics to the pavement mechanical properties. Using the FE model, the differences in the pavement mechanical behaviors resulting from the material isotropy, transverse isotropy, and orthotropy were compared and discussed. Furthermore, the need for treating pavement materials as orthotropic is addressed.

The remainder of this paper is organized as follows: Section 2 describes the methodology, including the experimental design, mathematical method, measurement of the anisotropy parameters, design of combinations, and finite element model. Section 3 presents the verification of the proposed model. Section 4 presents the analysis results, including the impact of anisotropy on surface deflection, impact of anisotropy on tensile stress, and impact of anisotropy on compressive strain. Section 5 presents the discussion of the results, and finally Section 6 presents the conclusions.

## 2. Methodology

### 2.1. Experimental Design

The overall experimental design of this study is depicted in Figure 1. As noted, A, B, and C represent material isotropy, transverse isotropy, and orthotropic, respectively. Moreover, each letter refers to the surface layer, base layer, and subgrade layer in order. For example, the ABA indicates a combination where the surface layer, base layer, and subgrade layer are respectively treated as isotropy, transverse isotropy, and isotropy.

First, we developed a 3-dimension (3D) FE model considering orthotropy. The orthotropic material parameters (i.e., the modulus ($E_x$, $E_y$, $E_z$, and $G_{yz}$) and Poisson's ratio ($\mu_{yz}$)), as critical inputs to the model, were calculated. This was achieved by proposing a new and efficient calculation method and the measurement of material parameters for each layer (HMA and semi-rigid base layers) by the material test system (MTS810 made in the USA).

Then, followed by detailed illustrations of the FE models, nine simulation scenarios (combinations of material parameters) are elaborated in-depth. Based on that, the contributions of each layer to the pavement mechanical behaviors are discussed thoroughly. Then, the FE models with experiment-based inputs are used to evaluate the anisotropy impacts on the pavement mechanical behaviors. Finally, two types of pavement structures are used to verify the feasibility of the developed model by using the falling weight deflectometer (FWD) data.

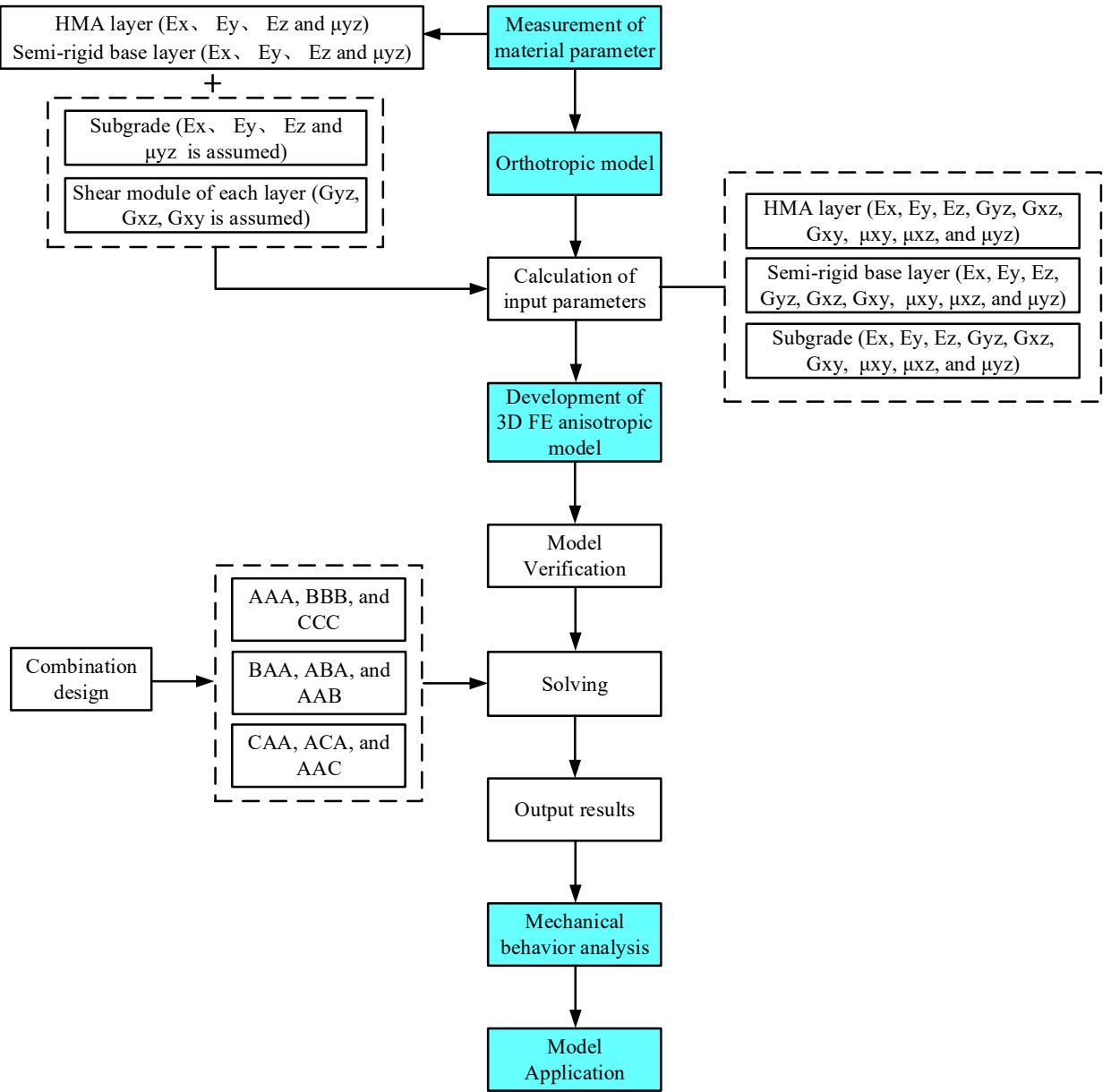

**Figure 1.** Our research methodology.

### 2.2. Mathematical Method

Materials with three mutually orthogonal elastic symmetrical surfaces are defined as orthotropic [32], and are composed of nine independent parameters ($E_x, E_y,$ and $E_z$; $\mu_{xy}, \mu_{xz},$ and $\mu_{yz}$; and $G_{yz}, G_{zx},$ and $G_{xy}$). The constitutive equation of orthotropic materials is given by

$$
\begin{Bmatrix} \varepsilon_x \\ \varepsilon_y \\ \varepsilon_z \\ \gamma_{yz} \\ \gamma_{zx} \\ \gamma_{xy} \end{Bmatrix} = \begin{bmatrix} \frac{1}{E_x} & \frac{-\mu_{yx}}{E_y} & \frac{-\mu_{zx}}{E_z} & 0 & 0 & 0 \\ \frac{-\mu_{xy}}{E_x} & \frac{1}{E_y} & \frac{-\mu_{zy}}{E_z} & 0 & 0 & 0 \\ \frac{-\mu_{xz}}{E_x} & \frac{-\mu_{yz}}{E_y} & \frac{1}{E_z} & 0 & 0 & 0 \\ 0 & 0 & 0 & \frac{1}{G_{yz}} & 0 & 0 \\ 0 & 0 & 0 & 0 & \frac{1}{G_{zx}} & 0 \\ 0 & 0 & 0 & 0 & 0 & \frac{1}{G_{xy}} \end{bmatrix} \begin{Bmatrix} \sigma_x \\ \sigma_y \\ \sigma_z \\ \tau_{yz} \\ \tau_{zx} \\ \tau_{xy} \end{Bmatrix}
\tag{1}
$$

where $\varepsilon_x, \varepsilon_y, \varepsilon_z, \gamma_{yz}, \gamma_{zx}$, and $\gamma_{xy}$ are the components of strains corresponding to the stresses $\sigma_x, \sigma_y, \sigma_z, \tau_{yz}, \tau_{zx}$, and $\tau_{xy}$. $E_x$, $E_y$, and $E_z$ are the moduli of elasticity in the transverse direction, longitudinal direction, and vertical directions, respectively; $G_{yz}, G_{zx}$, and $G_{xy}$ are the elastic shear modulus transverse, longitudinal, and vertical directions, respectively; $\mu_{xy}$, $\mu_{xz}$, and $\mu_{yz}$ are the major Poisson's ratios (PR) in the vertical, transverse, and longitudinal directions, respectively; and $\mu_{yx}$, $\mu_{zx}$, and $\mu_{zy}$ are the minor Poisson's ratios (NU) in the vertical, transverse, and longitudinal directions, respectively.

The relationship between the parameters of the orthotropic body is given by

$$\frac{\mu_{yx}}{E_y} = \frac{\mu_{xy}}{E_x}, \; \frac{\mu_{zx}}{E_z} = \frac{\mu_{xz}}{E_x}, \; \frac{\mu_{zy}}{E_z} = \frac{\mu_{yz}}{E_y} \tag{2}$$

The degree of anisotropy represents the ratio of material modulus or Poisson's ratio in any two directions of $x$, $y$, and $z$. Therefore, $k_1$ is defined as the anisotropy degree between the longitudinal direction and vertical direction. $k_2$ is defined as the anisotropy degree between the transverse direction and longitudinal direction. If $k_1$ and $k_2$ are equal to 1.0, the material is isotropic [26].

The anisotropy degree for the elastic modulus, shear modulus, and Poisson's ratio are assigned as $k_E, k_G$, and $k_\mu$, respectively. The anisotropy degree of the elastic modulus, shear modulus, and Poisson's ratio is assumed as equated and changing synchronously [27]; thereby, the specific anisotropy degree is given by

$$k_{E1} = \frac{E_y}{E_z}, k_{E2} = \frac{E_x}{E_y} \tag{3}$$

$$k_{\mu 1} = \frac{\mu_{zx}}{\mu_{yx}}, k_{\mu 2} = \frac{\mu_{yz}}{\mu_{xz}} \tag{4}$$

$$k_{G1} = \frac{G_{yx}}{G_{zx}}, k_{G2} = \frac{G_{xz}}{G_{yz}} \tag{5}$$

Combining Equations (1) and (2), the relationship among orthotropic parameters is obtained as follows:

$$\mu_{xz} = \frac{\mu_{yz}}{k_2}, \; \mu_{xy} = \frac{\mu_{yz}}{k_1^2 \cdot k_2} \tag{6}$$

$$G_{xz} = k_2 \cdot G_{yz}, \; G_{xy} = k_1 \cdot k_2 \cdot G_{yz} \tag{7}$$

Therefore, material anisotropy in each layer can be represented by five independent parameters ($E_x$, $E_y$, $E_z$, $\mu_{yz}$, and $G_{yz}$). Equations (6) and (7) are also applicable to the parameter calculation of transversely isotropic materials. For the transverse isotropy material, $k_2$ in Equations (6) and (7) is equal to 1.0. Therefore, the relationship among transverse isotropic parameters is given by

$$\mu_{xz} = \mu_{yz} = \mu', \; \mu_{xy} = \frac{\mu'}{k_1^2} \tag{8}$$

$$G_{xz} = G_{yz} = G', \; G_{xy} = k_1 \cdot G' \tag{9}$$

### 2.3. Measurement of Anisotropy Parameters

Based on the above calculation method, the material parameter tests were carried out. The elastic modulus of the mixture was obtained by performing the uniaxial compression test of the asphalt mixture according to the specification [41].

The mathematical model involves major and minor Poisson's ratios. In this paper, the major Poisson's ratio (PR) was used, and that of the orthotropic materials was tested using the MTS. The cuboid sample ($4 \times 4 \times 8$ cm) was placed on the MTS loading platform in the testing process, as shown in Figure 2. The vertical deformation was measured using four dial indicators on the loading plate; the lateral deformation in two directions was

tested, respectively, by two pairs of dial indicators located at the surface center at a height of 4 cm from the bearing. The sum of the values recorded by the corresponding pair of dial indicators refers to the deformation in that direction. The strain equals the ratio of the deformation to the corresponding thickness of the section, and then the Poisson's ratio can be calculated according to [42].

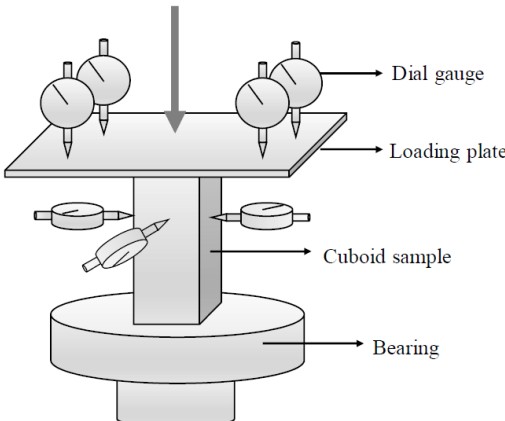

**Figure 2.** Test of the compression resilient modulus and Poisson's ratio of transverse isotropy materials.

Limited to the test conditions, the shear modulus $G_{yz}$ of each layer was first assumed [43], and the other shear modulus parameters were calculated according to Equation (4). As for subgrade, due to the enormous influence of the soil quality and particle size on the parameters, even if the parameters of a specific soil subgrade were measured, the representativeness was not satisfying. Thus, a typical value was adopted for the subgrade material parameters. The orthotropic parameters of pavement materials obtained from the test are shown in Table 1. The experimental results show that the anisotropy degrees $k_1$ and $k_2$ in the HMA layer were about 0.76 and 0.9, respectively. The anisotropy degrees $k_1$ and $k_2$ in the semi-rigid base layer were about 0.81 and 0.9, respectively.

**Table 1.** The five independent parameters of orthotropic materials.

| Pavement Structures | $E_x$ (MPa) | $E_y$ (MPa) | $E_z$ (MPa) | $\mu_{yz}$ | $G_{yz}$ (MPa) |
|---|---|---|---|---|---|
| Surface | 1267 | 1408 | 1668 | 0.18 | 628 |
| Semi-rigid base | 2592 | 2880 | 3200 | 0.18 | 1280 |
| Subgrade | 49 | 54 | 60 | 0.26 | 22 |

To facilitate the input parameter determination, software programmed by MAT-LAB was developed. The software interface was as shown in Figure 3. Using the five independent parameters ($E_x, E_y, E_z$, $\mu_{yz}$, and $G_{yz}$), the remaining four parameters ($G_{xz}, G_{xy}$, $\mu_{xz}$, and $\mu_{xy}$) are addressed.

### 2.4. Design of Combinations

To investigate the material orthotropy effects of each layer on the asphalt pavement mechanical behaviors, nine combinations of material properties were designed: three combinations of separated isotropy, transverse isotropy, and orthotropy and six combinations of transverse isotropy, orthotropy, and each layer. The nine combinations of material properties are shown in Table 2.

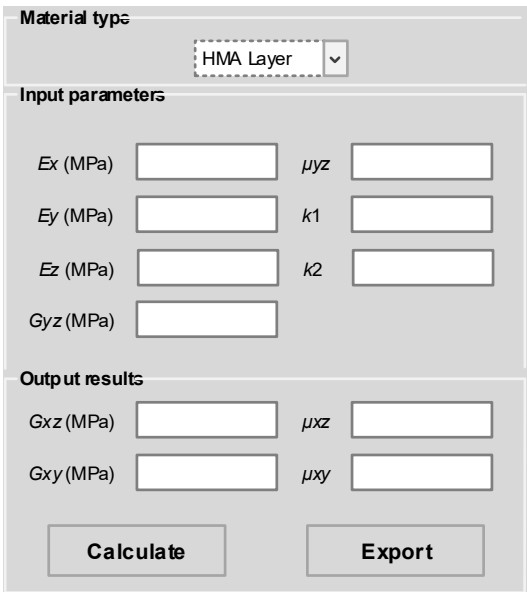

**Figure 3.** The material orthotropy estimation interface.

**Table 2.** The design of the material property combinations.

| No. | Surface | Semi-Rigid base | Subgrade |
| --- | --- | --- | --- |
| 1 | A | A | A |
| 2 | B | B | B |
| 3 | C | C | C |
| 4 | B | A | A |
| 5 | A | B | A |
| 6 | A | A | B |
| 7 | C | A | A |
| 8 | A | C | A |
| 9 | A | A | C |

Note: A, B, and C represent the isotropy, transversely isotropy, and orthotropy, respectively.

### 2.5. Finite Element Model

2.5.1. Pavement Structure

In this paper, a typical semi-rigid base asphalt pavement structure in China was adopted. The pavement structure was treated as a three-layer elastic continuous system, consisting of an 18-cm surface layer, a 54-cm semi-rigid base layer, and a filling subgrade layer. The surface irregularities during the operating phase have great effects on the asphalt pavement behaviors. In this paper, we assume that the surface irregularities are excellent to eliminate the influence of surface irregularities.

2.5.2. Geometry and Element

The 3D finite element model of anisotropic asphalt pavements is shown in Figure 4. The size of the FE model was determined as $10 \times 10 \times 16$ m (corresponding to the $x$, $y$, and $z$ three directions, respectively), and the coordinate origin was assigned at the center of the wheel gap. $x$, $y$, and $z$ represent the transverse, longitudinal, and vertical directions, respectively. The element type also has a significant influence on the calculation results. The solid 187 (10-node isoperimetric element) was selected as the computing element due to the flexibility of the anisotropic material properties.

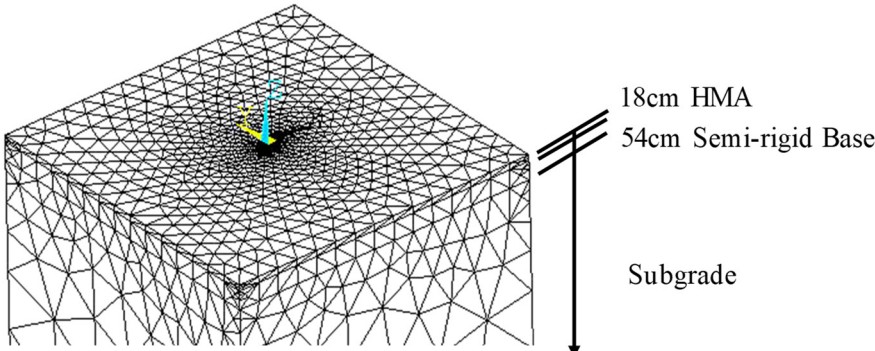

**Figure 4.** 3D Finite element model of the anisotropic asphalt pavement.

### 2.5.3. Meshing Method

The whole model was meshed first, and then local meshing refinement was performed on the loading area (e.g., the surface layer and semi-rigid base) to ensure the computation accuracy. The model after meshing can be found in Figure 4.

### 2.5.4. Boundary Condition

Displacement that was perpendicular to the direction $x$ or $y$ was bounded. At the bottom of the model, all direction displacement was bounded. The interlayer contacts were assumed as entirely continuous.

### 2.5.5. Load Model

The road contact areas of the tires and pressure distribution are essential factors for the load model. Tarefder and Ahmed [6] and Symes et al. [31] insisted that the tire ground imprint was not a uniform circular distribution, but closer to a rectangular distribution. For the conditions of different loads and tire pressures, there are three forms of wheel load distributions: uniform distribution, concave distribution, and convex distribution. With the rated load and certain tire pressure, the force distribution of the wheel on the road is approximately uniform. When the tire is overloaded or the tire pressure is insufficient, the wheel load on the road surface results in a concave distribution form, which is small in the middle and large on both sides.

When the tire is underloaded or the tire pressure is too high, the wheel force on the road results in a convex distribution form, more prominent in the middle and smaller on both sides [44]. Therefore, the non-uniform concave load corresponding to heavy traffic is more suitable to use as the load model in the paper, as shown in Figure 5a. Moreover, the heavy vehicle load is rather concave-distributed, and the maximum tire pressure is 1.217 MPa. In this paper, the FWD test data are used to verify the established FE model, and the wheel load subjected to pavement can be regarded as a sinusoidal load as shown in Figure 5b [45].

The longitudinal force on the pavement is remarkable, and it cannot be ignored. When the pavement suffers from a sizeable longitudinal force for an extended period, it is prone to rutting deformation. Given this, it is necessary to consider the longitudinal force in investigating the mechanical behaviors of asphalt pavements. The longitudinal force of the vehicle on the road is equal to the vertical wheel force multiplied by the vehicle-road adhesion coefficient. Due to the experimental limits related to the interaction between the tire and the pavement, the longitudinal force coefficient was taken in this study as 0.5 [46].

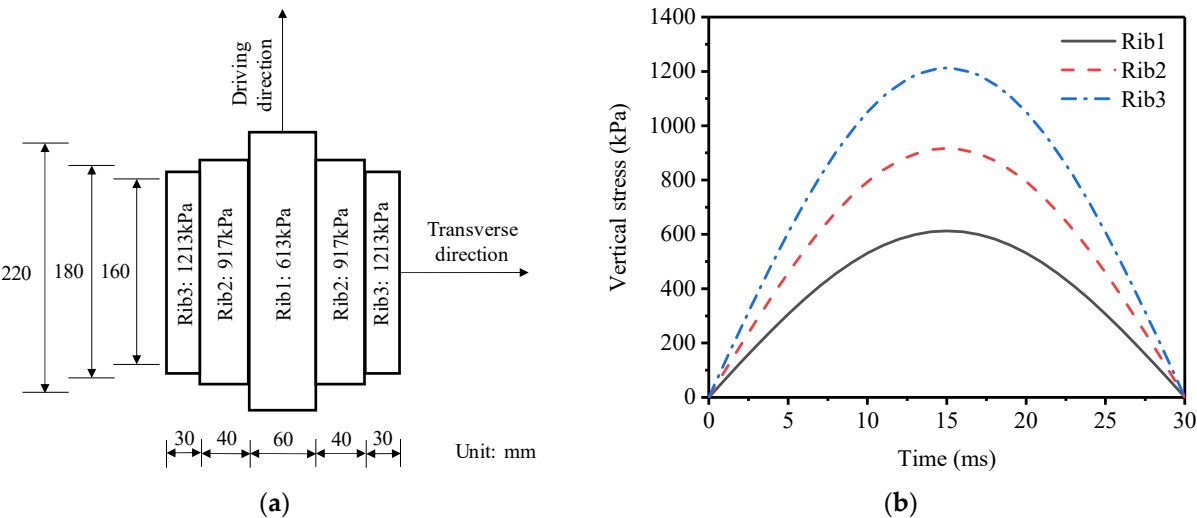

**Figure 5.** Load model in this paper: (**a**) Load distribution. (**b**) Vertical stress over time.

### 2.5.6. Mechanical Behavior Evaluation

In this study, the output indices, including the surface deflection, semi-rigid base bottom tensile stress, and compressive strain on top of the subgrade, were used to evaluate the bearing capacity, fatigue cracking, and rutting of pavement.

Considering the material anisotropy of each layer, the effect of anisotropy on the mechanical behaviors of asphalt pavement was analyzed. To allow direct comparisons, the changing rate of the pavement mechanical indices was defined as follows:

$$\Delta m = \frac{\delta_i - \delta_0}{\delta_0} \times 100\% \tag{10}$$

where $\Delta m$ is the changing rate of the mechanical index, dimensionless; and $\delta_0$ and $\delta_i$ are the computed values of the mechanical indices (e.g., stress, strain, deflection, etc.) corresponding to the material isotropy and anisotropy, respectively.

### 2.6. Statistical Analysis

Microsoft Excel 2016 was used for the statistical analysis of the data using the one-way analysis of variance (ANOVA). In this analysis, we assumed that the value of $\alpha$ was 0.05, indicating the probability of error in selecting the confidence coefficient. The determination of the inter-group error includes the ratio of the sum of squares to the degree of freedom, see Roman et al. [47]. When determining the significant effect of material anisotropy on the pavement mechanical behavior, the value of F is compared with the critical value of $F_{crit}$. If the value of F is greater than $F_{crit}$, the effect is significant; otherwise, it is not significant. In addition, If the *p*-value is less than 0.05, the influence is significant; otherwise, it is not significant. Therefore, the ANOVA methodology can help to determine the significant effect of the material anisotropy on the pavement mechanical behavior.

### 3. Model Verification

The established model was verified by comparing its results with the FWD data and using two typical types of pavement structure in southern China: semi-rigid and inversed pavements. Back-calculated moduli of the two types of pavements are used as the vertical moduli $E_z$ (Table 3). Based on the material parameters obtained in this paper, the anisotropy degrees $k_1$ of the surface layer, base layer, and subgrade layer were 0.75, 0.8, and 0.8, respectively, and the $k_2$ of the mentioned layer was 0.9 for all layers. According to Equations (6) and (7), the input parameters of the anisotropic material of each layer can be calculated, as shown in Table 3.

**Table 3.** The pavement structure and material parameters.

| Pavement Structure | Material Type | E (MPa) | μ | H (m) | Density (kg/m³) |
|---|---|---|---|---|---|
| Structure A | Surface | 1500 | 0.25 | 0.16 | 2400 |
| | 5% Semi-base | 2800 | 0.30 | 0.35 | 2300 |
| | 3% Semi-base | 2000 | 0.30 | 0.4 | 2300 |
| | Subgrade | 80 | 0.35 | - | 1800 |
| Structure B | Surface | 1500 | 0.25 | 10 | 2400 |
| | ATB-25semi-rigid base | 1800 | 0.25 | 16 | 2400 |
| | Granular base | 180 | 0.35 | 16 | 2300 |
| | Semi-base | 2000 | 0.30 | 32 | 2300 |
| | Subgrade | 80 | 0.35 | - | 1800 |

Note: $E$, $\mu$, and $h$ refer to the modulus, Poisson's ratio, and thickness of the material, respectively.

There were nine deflection sensors on the FWD, which were distributed in a straight line. The distance between each deflection sensor and the center of load action is shown in Table 4. To reduce the test error, each measuring point was hammered three times. The first two loads were mainly used to ensure the deflection bearing plate was closely in contact with the road surface and to eliminate the influence of residual loose particles on the measured deflection basin. The values obtained by the third load were recorded as valid data. The deflection basin results using the developed FE model were compared to the FWD data, as shown in Figure 6. The deflection at the distance $x = 0.15$ m of the developed FE model was compared to the FWD data in the location of the sensor number $d_1$. The difference in the surface defection changing rate between the field data and the FE model are shown in Figure 7.

**Table 4.** The distance from sensor to coordinate origin.

| Sensor Number | $d_1$ | $d_2$ | $d_3$ | $d_4$ | $d_5$ | $d_6$ | $d_7$ | $d_8$ | $d_9$ |
|---|---|---|---|---|---|---|---|---|---|
| Distance (m) | 0.15 | 0.35 | 0.45 | 0.60 | 0.75 | 1.05 | 1.35 | 1.65 | 1.95 |

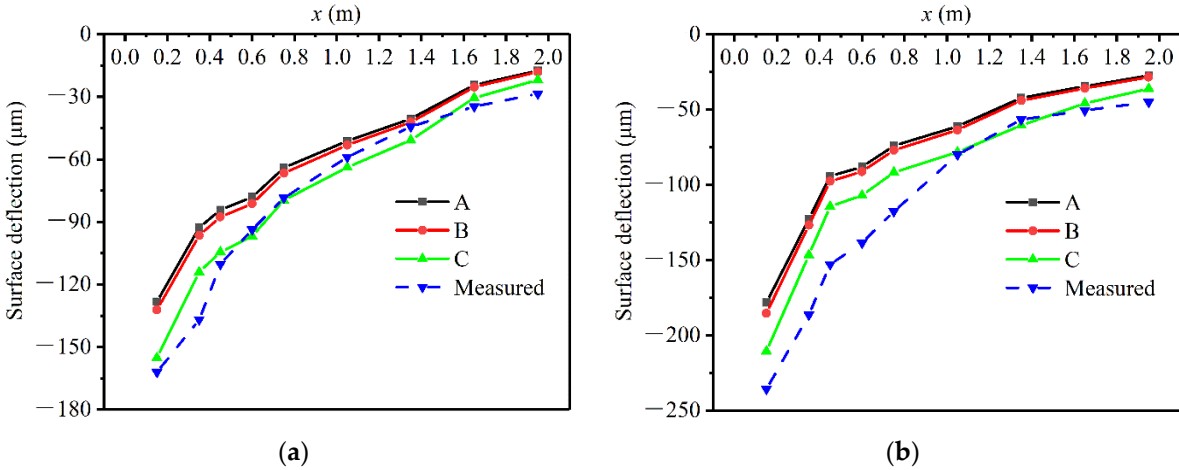

**Figure 6.** Comparison between the FWD data and anisotropy model of surface defection: (**a**) Structure A. (**b**) Structure B.

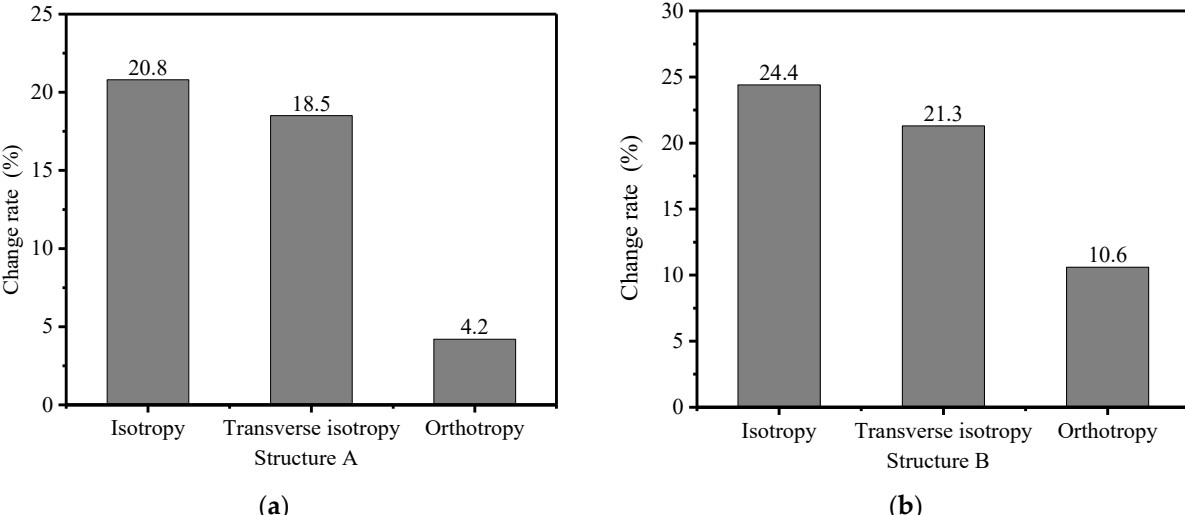

**Figure 7.** Change rate of the surface defection between the field data and the FE model: (**a**) Structure A. (**b**) Structure B.

As noted in Figure 6, almost all of the peak deflections obtained by the FE model were smaller than the measured values for the two pavement structures. In addition, in Figure 6a, the deflection-basin values of Structure A, a semi-rigid asphalt pavement, were close to the measured values, while those of Structure B in Figure 6b were not. This is because the base layer of Structure B was flexible, and thus more sensitive to the surrounding environment in the process of the field tests and, thereby, prone to generating larger surface deflections.

As noted, compared with the measured surface deflections, the change rates of the isotropy and transverse isotropy by the anisotropy FE model for Structure A were 20.8% and 18.5%, respectively, while the corresponding change rates for Structure B were 24.4% and 21.3%, respectively. The change rate of surface deflection of Structure A for the orthotropy was significantly reduced, at no more than 5% and that of Structure B for the orthotropy was 10.6%, showing a satisfying error level using the proposed anisotropy FE model. Therefore, the developed model can be used to predict pavement mechanical behavior effectively.

## 4. Analysis Results

### 4.1. Impact of Anisotropy on Surface Deflection

Surface deflection has been deemed to be an indicator of the pavement capacity performance, and an increased surface deflection corresponds to a lower capacity [5]. The distributions of surface deflection in the transverse direction under vertical and longitudinal loads are plotted in Figure 8.

The results of the ANOVA of pavement surface deflection are shown in Table 5. As noted, the significance $p$-value was less than 0.05, and the value of F was 12.401, which is larger than the critical value of 1.999, indicating that the material anisotropy had a significant effect on the road surface deflection.

As noted, the surface deflection increased first and then decreased in the transverse direction. The peak deflection occurred at $x = 0.15$ m, which corresponds to the center of the wheel loads. In Figure 8a, the surface deflection was the largest when considering the all-layer orthotropy, showing a peak value of 640 μm and a changing rate of 21.2%. Figure 8b,c implies that the surface deflection was slightly affected by the anisotropy in the HMA and base layers. Figure 8d demonstrates that the surface deflection for orthotropy in the subgrade was the largest, along with a value and changing rate of 605 μm and 9.0%, respectively, while those for transverse isotropy changed slightly.

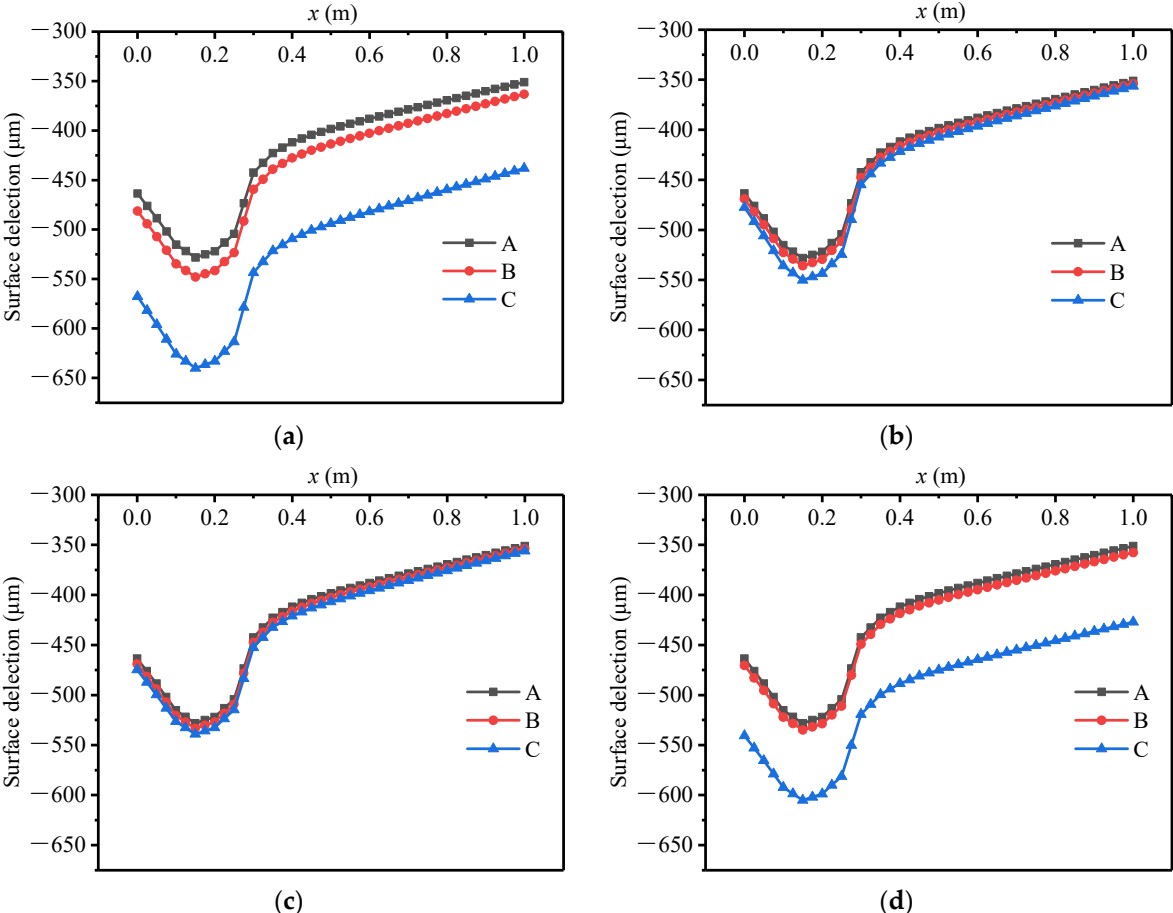

**Figure 8.** Surface deflections under the anisotropy of different materials: (**a**) Anisotropy in all layers. (**b**) Anisotropy in the HMA layer. (**c**) Anisotropy in the semi-rigid base layer. (**d**) Anisotropy in the subgrade layer.

**Table 5.** The results of the ANOVA of the surface deflection.

| Source of Difference | Sum of Squares | Degree of Freedom | Mean Square | F | *p*-Value | $F_{crit}$ |
|---|---|---|---|---|---|---|
| Between group | 201,098 | 8 | 25,137.25 | 12.401 | $1.29 \times 10^{-13}$ | 1.999 |
| Within group | 310,132.7 | 153 | 2027.011 | - | - | - |

We found that the material anisotropy increased the surface deflection to an extent, which was in good agreement with the results of Tarefder et al. [4]. Moreover, it should be noted that the subgrade orthotropy made the most significant contribution to the surface deflection under all of the investigated scenarios.

### 4.2. Impact of Anisotropy on Tensile Stress

Tensile stress at the bottom of the semi-rigid base layer is commonly applied to assess pavement fatigue performance [4]. Figure 9 depicts the changing rates of the lateral tensile stress at the bottom of the semi-rigid base layer under different combinations of anisotropic materials. The longitudinal tensile stress at the bottom of the semi-rigid base showed similar results. Therefore, the changing rates of the longitudinal tensile stress under different combinations of anisotropic materials are not presented in this paper.

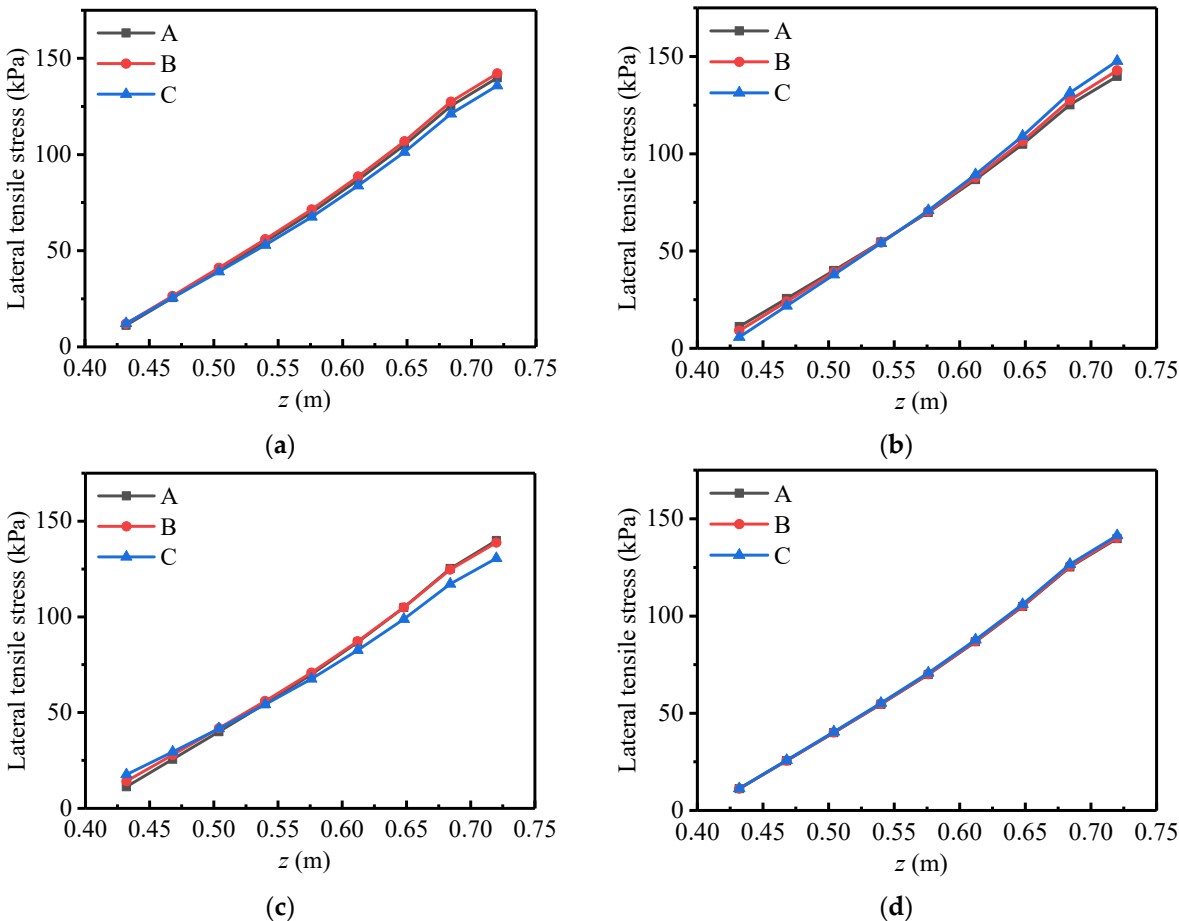

**Figure 9.** Lateral tensile stress under the anisotropy of different materials: (**a**) Anisotropy in all layers. (**b**) Anisotropy in the HMA layer. (**c**) Anisotropy in the semi-rigid base layer. (**d**) Anisotropy in the subgrade layer.

The results of the ANOVA of lateral tensile stress are shown in Table 6. As noted, the *p*-value of the lateral tensile stress at the bottom of the semi-rigid base layers is equal to 0.651 (greater than 0.05), and the F value is 0.746, which is less than the critical value of 2.070, indicating that the material anisotropy had no significant effect on the lateral tensile stresses at the bottom of the base.

**Table 6.** The results of the ANOVA of the lateral tensile stress.

| Source of Difference | Sum of Squares | Degree of Freedom | Mean Square | F | *p*-Value | F_crit |
|---|---|---|---|---|---|---|
| Between group | 22,303.57 | 8 | 2787.946 | 0.746 | 0.651 | 2.070 |
| Within group | 269,249.7 | 72 | 3739.579 | - | - | - |

Figure 9 shows that the all-layer anisotropy increased the lateral tensile stress at the bottom of the semi-rigid base, and that of the orthotropy was larger. The changing rate of the orthotropy in the HMA layer was the largest among all combinations at about 6%. Interestingly, the base layer anisotropy decreased the longitudinal tensile stress to some extent, about −2%. This is because the modulus of the anisotropic material in two directions on the horizontal plane was smaller than in the vertical direction. We found that the anisotropy in the HMA layer made the most significant positive contribution to the transverse tensile stress at the bottom of the base layer.

### 4.3. Impact of Anisotropy on Compressive Strain

Compressive strain can be used to show permanent deformation performance [20]. The compressive strain on the top of the subgrade is an essential mechanical index for permanent deformation. The results of the compressive strain on the subgrade top in the transverse direction are shown in Figure 10.

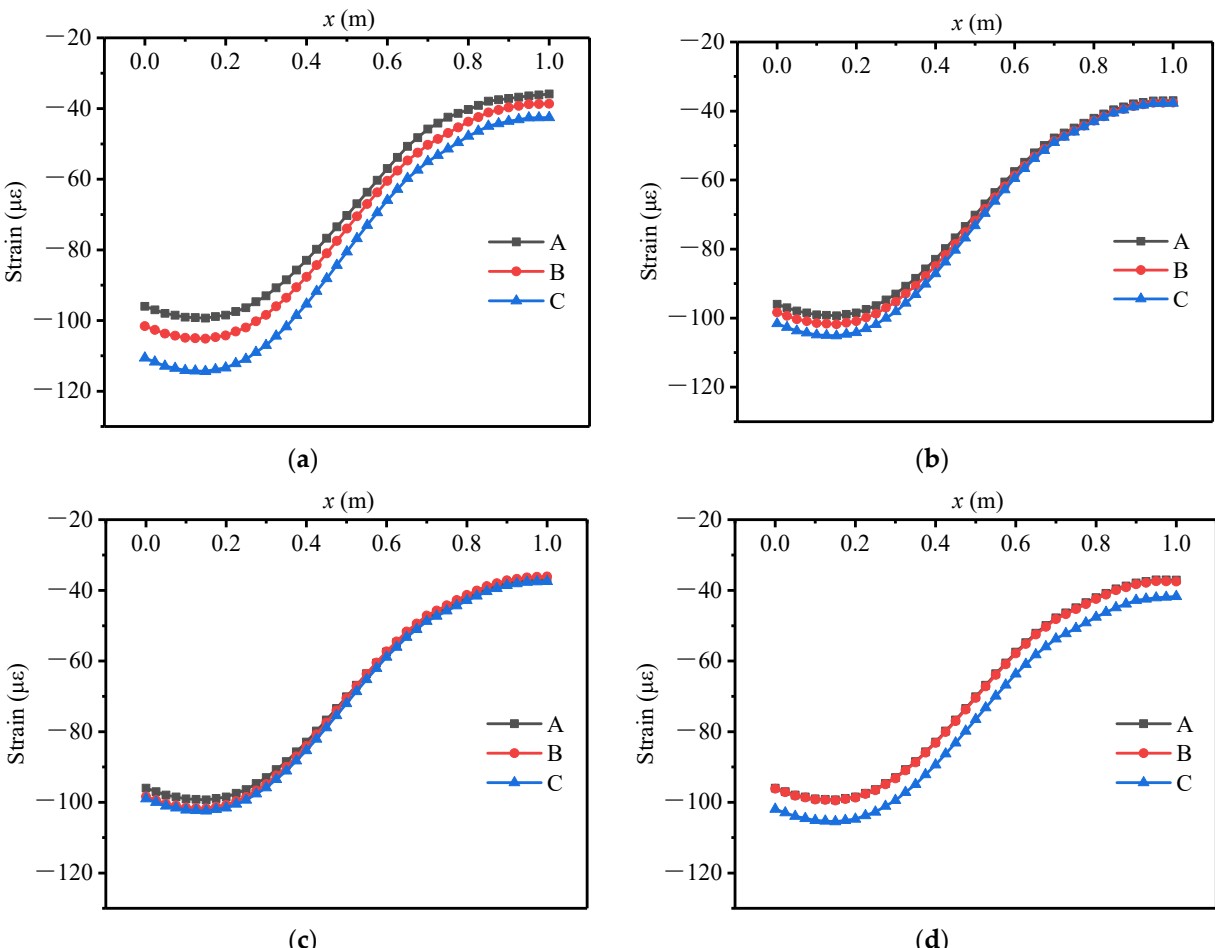

**Figure 10.** The effects of anisotropy of different materials on the compressive strain on the top of the subgrade: (**a**) Anisotropy in all layers. (**b**) Anisotropy in the HMA layer. (**c**) Anisotropy in the semi-rigid base layer. (**d**) Anisotropy in the subgrade layer.

The results of the ANOVA of compressive strain on top of the subgrade are shown in Table 7. As noted, the significant *p*-value of the compression strain at the subgrade top was less than 0.05, and the value of F was 8.346, which is larger than the critical value of 1.999, indicating that the material anisotropy had a significant effect on the compression strain at the subgrade top.

**Table 7.** The results of the ANOVA of the compressive strain on top of the subgrade.

| Source of Difference | Sum of Squares | Degree of Freedom | Mean Square | F | *p*-Value | $F_{crit}$ |
|---|---|---|---|---|---|---|
| Between group | 2722.112 | 8 | 340.264 | 8.346 | $2.3 \times 10^{-9}$ | 1.999 |
| Within group | 6238.086 | 153 | 40.772 | - | - | - |

Figure 10a shows that the absolute compressive strain on the subgrade top increased at a transverse distance of 0 to 0.15 m and decreased after that. As noted, the compressive strain on the subgrade top was remarkably affected by the orthotropy and transverse isotropy. In addition, the changing rate peak values for the all-layer orthotropy and transverse isotropy were 19.0% and 6.7%, respectively. Figure 10b,d show that the orthotropy in the HMA and subgrade layers had a larger effect on the compressive strain on top of the subgrade, with changing rates of 5.6% and 8.6%, respectively; while those of the transverse isotropy were relatively slight, 2.8% and 0.3%. In addition, the effects of the two anisotropies in the base layer are close, showing a changing rate of around 4% in Figure 10c.

A larger compressive strain was generated when the orthotropy was taken into consideration. Moreover, the subgrade layer orthotropy made the most significant contribution to the compressive strain under all of the investigated scenarios.

## 5. Discussion

Previous studies have shown that, with the increase in the anisotropy, the road surface deflection, tensile stress, and strain at the bottom of the layer increase, and the maximum change rate can reach more than 20% [5]. In addition, the maximum change rate of a mechanical index is related to the range of the anisotropy [26]. The larger the range, the greater the influence on the results. In this paper, the results showed that the orthotropy increased the surface deflection and compressive strain on the subgrade top by 21% and 19%, respectively, which is consistent with previous studies.

Based on the test results of the material parameters, the minimum anisotropy values of surface course and base course materials were 0.76 and 0.81, respectively, which are not less than 0.7. However, most of the material parameters used by previous scholars were based on assumed values, and the anisotropy could range from 0.17 to 1 [48]. The FE results in this study showed that: (1) the maximum deflection change rate of the pavement surface was 3.8% under the transverse isotropy, (2) the maximum change rate of the maximum tensile stress at the base was 3.0%, (3) the maximum change rate of the maximum compressive strain on the top of the subgrade was 6.7%, and (4) the absolute value of the maximum change rate of the three mechanical indexes was not more than 7%. Given this, the reason for the small change rate of the above mechanical indices was that the anisotropy range of material parameters was small in the FE calculations.

In addition, the effects of the transversely isotropic and orthotropic materials on the mechanical properties were identical in this study while their change rates of the material anisotropy effect on mechanical properties were different. By comparing the difference of the maximum change rate of each mechanical index between the two kinds of anisotropic materials, we can deeply understand the difference in their influence on the asphalt pavement performance. The more significant the difference between them, the more sensitive it is to reflect the difference of the two materials effect on the asphalt pavement mechanical properties. Three pavement mechanical indices were evaluated: the (1) maximum deflection of pavement surface, (2) maximum tensile stress at the bottom of pavement layer, and (3) maximum compressive strain on the subgrade top.

In the three indices, the difference in the maximum deflection change rate of pavement surface was the most prominent (17.4%), indicating that the difference between the transversely isotropic and orthotropic materials was the most sensitive to the pavement bearing capacity. The difference in the change rate of the maximum compressive strain on the subgrade top was 12.3%, indicating that the difference in the properties of the transversely isotropic and orthotropic materials was more sensitive to the maximum compressive strain on the subgrade top.

The difference in the change rate of tensile stress at the bottom of the pavement layer was the most minuscule (−7.6%), indicating that the difference between the two materials was the least sensitive to the maximum tensile stress at the bottom of the pavement layer. The comparison results showed that considering only the material transverse isotropy was

insufficient. This may underestimate the influence of material anisotropy on the mechanical properties of pavement. Thus, it is necessary to consider the material orthotropy further.

Based on the influence of orthotropic materials on the mechanical behavior of pavement, three significant suggestions for controlling the material during the process of pavement construction are put forward, as follows:

(1) The increased surface deflection illustrates that those materials (in all layers or each layer) only considered as isotropic are unfavorable for the pavement bearing capacity. Therefore, we recommend that the homogeneity degree (especially for the filling subgrade) be strictly controlled during construction to assure adequate pavement capacity.

(2) The tensile stress illustrates that pavement fatigue performance might be underestimated or over-estimated if the HMA layer or base layer, respectively, is only considered as isotropic. Therefore, we recommend that the homogeneity degree of the surface layer be strictly controlled during construction to assure pavement fatigue performance. In contrast, the base layer can be treated as usual.

(3) The increased compressive strain illustrates that the permanent deformation might be underestimated if the materials (in all layers or each layer) are only considered isotropic. Therefore, we recommend that the homogeneity degree (especially for the filling subgrade) be strictly controlled during construction to assure the pavement anti-rutting performance.

## 6. Concluding Remarks

To thoroughly evaluate the impacts of material orthotropy on pavement mechanical behavior, this paper presented a new and efficient 3D FE model of anisotropic materials. Based on this model, nine combinations of material characteristics were designed to determine the contributions of each layer orthotropy to the pavement mechanical properties. Finally, the anisotropic FE model with experiment-based inputs was applied and compared with field measurements. Based on this study, the following comments are offered:

- The proposed 3D FE model that considers material orthotropy can investigate the anisotropy impacts on pavement mechanical behaviors. The results showed that these impacts should not be ignored and that the all-layer orthotropy was the most unfavorable combination. The results also showed that the material orthotropy had larger impacts on the surface deflection than did the transverse isotropy.

- The anisotropy degrees of the HMA and semi-rigid base layers were determined experimentally. The results can be used as a basis for inputs to analyze pavement mechanical behavior considering the material anisotropy.

- Compared with the isotropic material, the orthotropy significantly increased the surface deflection and the compressive strain on top of the subgrade by 21% and 19%, respectively. The orthotropy impacts in the HMA layer were noticeable in the tensile stress at the bottom of the base layer, while the impacts on the surface deflection were slight. The subgrade orthotropy made the most significant contributions to the surface deflection and the compressive strain on top of the subgrade. Orthotropy in the base layer had slight effects on the mechanical behaviors.

- The developed 3D FE model was shown to be in good agreement with the FWD data of semi-rigid asphalt pavement, showing that this model can be used to predict the mechanical behaviors of semi-rigid asphalt pavements. The proposed 3D FE model, which considers the orthotropic characteristics of pavement materials, provides a practical measure for the mechanical behavior analysis of asphalt pavements. We recommend that the homogeneity degree of the filling subgrade should be strictly controlled to assure adequate pavement capacity and anti-rutting performance during construction.

- The material anisotropy determined in this paper was based on the typical semi-rigid pavement structure. In the future, material parameters under different conditions (e.g., temperatures, moisture, and load) should be tested to provide more accurate in-

puts for the proposed 3D FE model. Finally, the most unfavorable anisotropy combination should be determined considering different pavement structures (e.g., inverted and composite).

**Author Contributions:** Conceptualization, M.L.; Formal analysis, M.L.; Methodology, M.L. and Z.J.; Project administration, H.G.; Software, M.L.; Supervision, C.H., H.G., and S.M.E.; Validation, M.L.; Writing—original draft, M.L.; Writing—review and editing, S.M.E. All authors have read and agreed to the published version of the manuscript.

**Funding:** This research was funded by the National Natural Science Foundation of China, grant number 52078062 and the Natural Science Foundation of Hunan Province, grant number 2020JJ4604.

**Institutional Review Board Statement:** Not applicable.

**Informed Consent Statement:** Not applicable.

**Data Availability Statement:** Data sharing is not applicable to this article.

**Acknowledgments:** The authors are grateful to three anonymous reviewers for their constructive suggestions and comments that have improved the content and presentation of the paper.

**Conflicts of Interest:** The authors declare no conflict of interest.

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
