# Peer review of "Impacts of Material Orthotropy on Mechanical Behaviors of Asphalt Pavements"

_applsci, doi:10.3390/app11125481_

Round 1

Reviewer 1 Report

The paper investigates the influence of the anisotropy of road pavement layers in the evaluation of their performances.

The study could be improved describing in more details the load model adopted, highlighting the limits of the experimentation related to the interaction between the tire and the pavement.

Furthermore, they should comment or include in the experimentation the analysis of the expected surface irregularities that characterize the construction phase, also adopting an excellent construction process. It is assumed that these irregularities, combined with the phenomenon of fatigue, can have during the operating phase greater effects than those investigated, that in any case remain interesting.

A typo in fig. 9

Author Response

Response to Reviewer 1:

  1. The paper investigates the influence of the anisotropy of road pavement layers in the evaluation of their performances. The study could be improved describing in more details the load model adopted, highlighting the limits of the experimentation related to the interaction between the tire and the pavement.

Response: We revised the details of the load model and highlighted the limitations of the experimentation related to the tire and the pavement interactions, as follows (Lines 241- 263):

“2.5.5 Load Model

The road contact areas of the tires and pressure distribution are essential factors for the load model. Tarefder and Ahmed [6,32] insisted that the tire ground imprint was not uniform circular distribution, but much closer to the rectangular distribution. For the conditions of different load and tire pressure, there are 3 forms of wheel loads distribution, which is uniform distribution, concave distribution and convex distribution. With the rated load and certain tire pressure, the force distribution of the wheel on the road is approximately uniform; When the tire is overloaded or the tire pressure is insufficient, the force of the wheel on the road surface presents a concave distribution form, which is small in the middle and large on both sides; When the tire is under loaded or the tire pressure is too high, the force of the wheel on the road presents a convex distribution form, which is larger in the middle and smaller on both sides[45]. Therefore, the non-uniform concave load corresponding to heavy traffic is more suitable as the load model in the paper, as shown in Figure 5(a). Moreover, the heavy vehicle load is rather concave-distributed and the maximum tire pressure is 1.217MPa. In this paper, the FWD test data is used to verify the established FE model and the wheel load subjected to pavement can be regarded as a sinusoidal load as shown in Figure 5(b) [46].

The longitudinal force subjected to the pavement is remarkable, and it cannot be ignored. When pavement is suffered from the sizeable longitudinal force for a long period, it is prone to rutting deformation. In view of this, it is necessary to consider the longitudinal force in investigating mechanical behaviors of asphalt pavements. The longitudinal force of the vehicle on the road is equal to the vertical wheel force multiplied by the vehicle-road adhesion coefficient. Due to the limits of the experimentation related to the interaction between the tire and the pavement, the coefficient of longitudinal force is supposed and taken as 0.5 in this study [47].

  1. Furthermore, they should comment or include in the experimentation the analysis of the expected surface irregularities that characterize the construction phase, also adopting an excellent construction process. It is assumed that these irregularities, combined with the phenomenon of fatigue, can have during the operating phase greater effects than those investigated, that in any case remain interesting.

Response: In this paper, we assumed that surface irregularities are excellent and have less effects on asphalt pavement behaviors. This assumption was clarified in the revised paper, as follows (Lines 219-221):

“The surface irregularities have during the operating phase great effects on asphalt pavement behaviors. In this paper, we assume that surface irregularities are excellent to eliminate the influence of surface irregularities.”

  1. A typo in fig. 9

Response: We have corrected the typo in Fig. 6.

Reviewer 2 Report

Manuscript incorrectly prepared in terms of editing.

The manuscript is a little short, but broken down over many pages.

The discussion was not conducted in accordance with the guidelines of journal.

There is no in-depth statistical analysis.

An expansion of literature, conclusions and discussions is required.

In this form, I advise against publishing the article in this journal.

English needs a lot of improvement.

Author Response

Response to Reviewer 2:

  1. Manuscript incorrectly prepared in terms of editing.

Response: We downloaded the latest papers from official website for reference and modified the manuscript carefully according to the “MDPI Style Guide”.

  1. The manuscript is a little short, but broken down over many pages.

Response: We expanded the following sections: Introduction, Load Model, Results, and Conclusions.

  1. The discussion was not conducted in accordance with the guidelines of journal.

Response: We adjusted the discussion section in accordance with the guidelines of the journal. A separate discussion section was added, as follows (Lines 375-428):

“5. Discussion

Previous studies have shown that with the increase in the anisotropy, road surface deflection, tensile stress, and strain at the bottom of the layer increase, and the maximum change rate can reach more than 20% [5]. In addition, the maximum change rate of a mechanical index is related to the range of the anisotropy [27]. The larger the range, the greater the influence on the results. In this paper, the results showed that the orthotropy increased surface deflection and compressive strain on the subgrade top by 21% and 19%, respectively, which is consistent with previous studies.

Based on the test results of material parameters, the minimum anisotropy of surface course and base course materials is 0.76 and 0.81, respectively, which are not less than 0.7. However, most of the material parameters used by previous scholars were based on assumed values, and the anisotropy could range from 0.17 to 1 [48]. The FE results in this study showed that: (1) the maximum deflection change rate of the pavement surface was 3.8% under the transverse isotropy, (2) the maximum change rate of the maximum tensile stress at the base was 3.0%, (3) the maximum change rate of the maximum compressive strain on the top of the subgrade was 6.7%, and (4) the absolute value of the maximum change rate of the three mechanical indexes was not more than 7%. Given this, the reason for the small change rate of the above mechanical indices was that the anisotropy range of material parameters was small in the FE calculations.

In addition, the effects of the transversely isotropic and orthotropic materials on the mechanical properties were identical in this study while their change rates of the material anisotropy effect on mechanical properties were different. By comparing the difference of the maximum change rate of each mechanical index between the two kinds of anisotropic materials, we can deeply understand the difference in their influence on asphalt pavement performance. The more significant the difference between them, the more sensitive it is to reflect the difference of the two materials effect on the asphalt pavement mechanical properties. Three pavement mechanical indices were evaluated: (1) maximum deflection of pavement surface, (2) maximum tensile stress at the bottom of pavement layer, and (3) maximum compressive strain on the subgrade top.

In the three indices, the difference in the maximum deflection change rate of pavement surface was the most prominent (17.4%), indicating that the difference between the transversely isotropic and orthotropic materials was most sensitive to pavement bearing capacity. The difference in the change rate of the maximum compressive strain on the subgrade top was 12.3%, indicating that the difference in the properties of the transversely isotropic and orthotropic materials was more sensitive to the maximum compressive strain on the subgrade top. The difference in the change rate of tensile stress at the bottom of the pavement layer was the most minuscule (-7.6%), indicating that the difference between the two materials was less sensitive to the maximum tensile stress at the bottom of the pavement layer. The comparison results showed that considering only material transverse isotropy was not enough. This may underestimate the influence of material anisotropy on the mechanical properties of pavement. Thus, it is necessary to consider material orthotropy further.

Based on the influence of orthotropic materials on the mechanical behavior of pavement, three significant suggestions for controlling the material during the process of pavement construction are put forward, as follows:

  • The increased surface deflection illustrates that those materials (in all layers or each layer) only considered as isotropic are unfavorable for the pavement bearing capacity. Therefore, it is recommended that the homogeneity degree (especially for the filling subgrade) be strictly controlled during construction to assure adequate pavement capacity.
  • The tensile stress illustrates that pavement fatigue performance might be under-estimated (or over-estimated) when the HMA layer (or base layer) was only considered as isotropic, respectively. Therefore, it is recommended that the homogeneity degree of the surface layer be strictly controlled during construction to assure pavement fatigue performance. In contrast, that for the base layer can be treated as usual.
  • The increased compressive strain illustrates that permanent deformation might be underestimated when the materials (in all layers or each layer) were only considered isotropic. Therefore, it is recommended that the homogeneity degree (especially for the filling subgrade) be strictly controlled during construction to assure the pavement anti-rutting performance.”

  1. There is no in-depth statistical analysis.

Response: We added in-depth statistical analysis in Table 5 and Table 6 in the Analysis Results section, as follows:

Table 5 Maximum change rate of surface deflection considering material anisotropy

Maximum change rate of surface deflection (%)

Anisotropy in all layer

Anisotropy in HMA layer

Anisotropy in semi-rigid base layer

Anisotropy in subgrade

Transverse isotropy

3.8

1.8

1.3

0.8

Orthotropy

21.2

3.1

2.6

9.0

Table 6 Maximum change rate of compressive strain on top of subgrade considering material anisotropy

Maximum change rate of compressive strain on top of subgrade (%)

Anisotropy in all layer

Anisotropy in HMA layer

Anisotropy in semi-rigid base layer

Anisotropy in subgrade

Transverse isotropy

6.7

2.8

3.8

0.3

Orthotropy

19.0

5.6

4.8

8.6

  1. An expansion of literature, conclusions and discussions is required.

Response: We have expanded the literature (which is now part of the Introduction), Load Model, and Conclusions and Discussions.

  1. In this form, I advise against publishing the article in this journal. English needs a lot of improvement.

Response: We made great efforts to improve the manuscript, as can be found in the revised version and the English writing of the paper was significantly polished by a native English professor.

Reviewer 3 Report

  • How many words has your manuscript?
  • It is not a good idea to use acronyms in the abstract. You should at least indicate the meaning when you use it for the first time. For example FWD
  • The abstract should be like a short story. Your abstract is hard to read.
  • I think you should merge the introduction and literature review. 
  • You need to clearly state your objective. Lines 59-73 are not really objectives, but more like scope. State clearly what problem you want to solve.
  • Your introduction and literature review should include some more general introduction to guide readers to your more specific information. You may include some general information about mechanical behaviors of asphalt pavements such as doi.org/10.1155/2018/1650945 or performance of asphalt mixtures such as doi.org/10.1080/14680629.2021.1908408
  • The formatting of your paper is different than other papers I reviewed for Applied Sciences. You should check formatting, especially in-text citations.
  • You need to improve the “results and discussion” section. Especially discussion part. I can`t see you analyzing in depth your results. The biggest weakness of your manuscript is the language. Please, make it easier to read.

Author Response

  1. How many words has your manuscript?

Response: 7194 words.

  1. It is not a good idea to use acronyms in the abstract. You should at least indicate the meaning when you use it for the first time. For example FWD

Response: The full name of the acronyms in the abstract and the manuscript have been defined when they were mentioned for the first time.

  1. The abstract should be like a short story. Your abstract is hard to read.

Response: The abstract was revised and shortened to be easy to read.

  1. I think you should merge the introduction and literature review.

Response: We have merged the introduction and literature review.

  1. You need to clearly state your objective. Lines 59-73 are not really objectives, but more like scope. State clearly what problem you want to solve.

Response: The content you mentioned was actually not the objectives of this work. Instead, it was the contributions. We have rewritten the contributions clearly, as follows (Lines 97-107):

“(1) A new and efficient 3D FE model of the anisotropic pavement material considering the consistency of the parameters (e.g., elastic modulus, shear modulus, and Poisson's ratio) is developed, which derives the relationship among anisotropic parameters.

(2) Based on the degree of anisotropy and considering parameters’ consistency, the anisotropic model’s independent parameters were reduced from 9 to 5, which is fairly convenient for determining the input parameters.

(3) Nine different combinations of material characteristics that considered the contributions of each layer of material characteristics to the pavement mechanical properties were designed. Using the FE model, the differences in pavement mechanical behaviors resulting from material isotropy, transverse isotropy, and orthotropy were compared and discussed. Furthermore, the need for treating pavement materials as orthotropic was addressed.”

The objectives of this manuscripts are shown, as follows (Lines 93-95):

“Specifically, this paper aims to address two questions: What are the parameter relationships of the orthotropic body considering the consistency of the parameters? and What are the impacts of material orthotropy on mechanical behaviors of asphalt pavement?”

  1. Your introduction and literature review should include some more general introduction to guide readers to your more specific information. You may include some general information about mechanical behaviors of asphalt pavements such as doi.org/10.1155/2018/1650945 or performance of asphalt mixtures such as doi.org/10.1080/14680629.2021.1908408

Response: The two references mentioned above are included in the revised paper. We read them carefully and added some general information about mechanical behaviors of asphalt pavements considering material transverse isotropy in the Introduction section, as shown below (Lines 82-85):

“In particular, with the help of commercial software (e.g., SAP, NASTRAN, ANSYS, and ABQUES), the pavement mechanical behaviors [40, 41] (e.g., deformation, stress, and strain) at any position can be obtained easily.”

References:

[40] Baldo N., Manthos E., Pasetto M. Analysis of the Mechanical Behaviour of Asphalt Concretes Using Artificial Neural Networks. Advances in Civil Engineering, 2018, 2018(PT.5):1650945.1-1650945.17.

[41] Polaczyk P., Ma Y., Xiao R., Hu W., Jiang X., & Huang B. S. Characterization of aggregate interlocking in hot mix asphalt by mechanistic performance tests. Road Materials and Pavement Design, 2021.

  1. The formatting of your paper is different than other papers I reviewed for Applied Sciences. You should check formatting, especially in-text citations.

Response: We have modified the in-text citations and the reference format.

  1. You need to improve the “results and discussion” section. Especially discussion part. I can`t see you analyzing in depth your results.

Response: We adjusted the discussion section in accordance with the guidelines of the journal. A separate discussion section was added, as follows (Lines 375-428):

“5. Discussion

Previous studies have shown that with the increase in the anisotropy, road surface deflection, tensile stress, and strain at the bottom of the layer increase, and the maximum change rate can reach more than 20% [4-6]. In addition, the maximum change rate of a mechanical index is related to the range of the anisotropy. The larger the range, the greater the influence on the results. The results showed that the orthotropy increased surface deflection and compressive strain on the subgrade top by 21% and 19%, respectively, which is consistent with previous studies.

Based on previous test results of material parameters, the minimum anisotropy of surface course and base course materials is 0.76 and 0.81, respectively, which are not less than 0.7 [27]. However, most of the material parameters used by previous scholars were based on assumed values, and the anisotropy could range from 0.17 to 1 [22, 28]. The previous FE results showed that: (1) the maximum deflection change rate of the pavement surface was 3.8% under the transverse isotropy, (2) the maximum change rate of the maximum tensile stress at the base was 3.0%, (3) the maximum change rate of the maximum compressive strain on the top of the subgrade was 6.7%, and (4) the absolute value of the maximum change rate of the three mechanical indexes was not more than 7%. Given this, the reason for the small change rate of the above mechanical indices was that the anisotropy range of the measured material parameters was small in the FE calculations.

In addition, the effects of the transversely isotropic and orthotropic materials on the mechanical properties were identical. By comparing the difference of the maximum change rate of each mechanical index between the two kinds of anisotropic materials, we can deeply understand the difference in their influence on asphalt pavement performance. The more significant the difference between them, the more sensitive it is to reflect the influence of the two materials on the mechanical properties of asphalt pavement. Three pavement mechanical indices were evaluated: (1) maximum deflection of pavement surface, (2) maximum tensile stress at the bottom of pavement layer, and (3) maximum compressive strain on the subgrade top.

In the three indices, the difference in the maximum deflection change rate of pavement surface was the most prominent (21.2%), indicating that the difference between the transversely isotropic and orthotropic materials was most sensitive to pavement bearing capacity. The difference in the change rate of the maximum compressive strain on the subgrade top was 19%, indicating that the difference in the properties of the transversely isotropic and orthotropic materials was more sensitive to the maximum compressive strain on the subgrade top. The difference in the change rate of tensile stress at the bottom of the pavement layer was the most minuscule (-8.5%), indicating that the difference between the two materials was less sensitive to the maximum tensile stress at the bottom of the pavement layer. The comparison results showed that considering only material transverse isotropy was not enough. This may underestimate the influence of material anisotropy on the mechanical properties of pavement. Thus, it is necessary to consider material orthotropy further.

Based on the results of the previous sections, when considering orthotropy, the following essential findings are drawn:

  • The increased surface deflection illustrates that those materials only considered as isotropic are unfavorable for the pavement bearing capacity. Therefore, it is recommended that the homogeneity degree (especially for the filling subgrade) be strictly controlled during construction to assure adequate pavement capacity.
  • The tensile stress illustrates that pavement fatigue performance might be under-estimated (or over-estimated) when the HMA layer (or base layer) was only considered as isotropic, respectively. Therefore, it is recommended that the homogeneity degree of the surface layer be strictly controlled during construction to assure pavement fatigue performance. In contrast, that for the base layer can be treated as usual.
  • The high peak compressive strain illustrates that permanent deformation might be underestimated when the materials were only considered isotropic. Therefore, it is recommended that the homogeneity degree (especially for the filling subgrade) be strictly controlled during construction to assure the pavement anti-rutting performance.”

  1. The biggest weakness of your manuscript is the language. Please, make it easier to read.

Response: The paper was carefully edited and polished to make it easier to read.

Round 2

Reviewer 2 Report

The article has not been properly corrected and still contains errors.
In figure 6 we can see the percentage means but no statistical analysis and description of these values can be seen. the chapter ends with an indescribable graph.
Figure 8, also no statistics. How many repetitions were there? There are no dependency analyzes.
This applies not only to figures but also to tables.
The ANOVA methodology has been included in the article below, from which I propose to take this information and refer to it:

Roman, K.; Roman, M.; Szadkowska, D.; Szadkowski, J.; Grzegorzewska, E. Evaluation of Physical and Chemical Parameters According to Energetic Willow (Salix viminalis L.) Cultivation. Energies 2021, 14, 2968. https://doi.org/10.3390/en14102968

In those articles, you can also find information about correlation using ANOVA with the Duncan test. Similar studies, in particular the division into homogeneous groups, were also missing in the article.

Only Chinese authors are cited in the references. How does this relate to research in the world?
The manuscript should not be published in this form 

Reviewer 3 Report

Thank you for addressing my comments.

Author Response

  1. Thank you for addressing my comments.

Response: Thank you.

Round 3

Reviewer 2 Report

Manuscript was improved correctly.